# Physical and Chemical Properties of Waste Foundry Exhaust Sand for Use in Self-Compacting Concrete

**DOI:** 10.3390/ma14195629

**Published:** 2021-09-28

**Authors:** Maria Auxiliadora de Barros Martins, Lucas Ramon Roque da Silva, Maria Gabriela A. Ranieri, Regina Mambeli Barros, Valquíria Claret dos Santos, Paulo César Gonçalves, Márcia Regina Baldissera Rodrigues, Rosa Cristina Cecche Lintz, Luísa Andréa Gachet, Carlos Barreira Martinez, Mirian de Lourdes Noronha Motta Melo

**Affiliations:** 1Physical and Chemical Institute, Federal University of Itajubá, BPS Avenue, 1303, Itajubá 37500-903, MG, Brazil; deiamabmartins@gmail.com; 2Mechanical Engineering Institute, Federal University of Itajubá, BPS Avenue, 1303, Itajubá 37500-903, MG, Brazil; lucasramonroque@gmail.com (L.R.R.d.S.); cmartinez@unifei.edu.br (C.B.M.); mirianmottamelo@unifei.edu.br (M.d.L.N.M.M.); 3Development, Technology and Society Program, Institute of Production Engineering and Management, Federal University of Itajubá, BPS Avenue, 1303, Itajubá 37500-903, MG, Brazil; 4Natural Resources Institute, Federal University of Itajubá, BPS Avenue, 1303, Itajubá 37500-903, MG, Brazil; mambeli@unifei.edu.br (R.M.B.); valquiria@unifei.edu.br (V.C.d.S.); paulocg9@unifei.edu.br (P.C.G.); mbaldissera@unifei.edu.br (M.R.B.R.); 5School of Technology, University of Campinas, Paschoal Marmo, 1888, Campinas 18484-332, SP, Brazil; rosalint@unicamp.br (R.C.C.L.); gachet@unicamp.br (L.A.G.)

**Keywords:** foundry exhaust sand, waste foundry sand, microstructural characterization, self-compacting concrete

## Abstract

The reuse of waste in civil construction brings environmental and economic benefits. However, for these to be used in concrete, it is necessary a previous evaluation of their physical and chemical characteristics. Thus, this study aimed to characterize and analyze the waste foundry exhaust sand (WFES) for use in self-compacting concrete (SCC). Foundry exhaust sand originates from the manufacturing process of sand molds and during demolding of metal parts. It is a fine sand rich in silica in the form of quartz collected by baghouse filter. Characterization of WFES was conducted through laser granulometry, scanning electron microscopy (SEM) in the energy dispersive spectroscopy (EDS) mode, X-ray diffraction (XRD), X-ray fluorescence (XRF), Fourier transform infrared spectroscopy (FTIR), thermogravimetry (TG) and derivative thermogravimetry (DTG) techniques. The waste was classified as non-hazardous and non-inert, with physical and chemical properties suitable for use in SCC composition, as fine aggregate or mineral addition. Five mixtures of SCC were developed, in order to determine the waste influence in both fresh and hardened concrete. The properties in the fresh state were reached. There was an increase in compressive strength and sulfate resistance, a decrease in water absorption of self-compacting concrete by incorporating WFES as 30% replacement.

## 1. Introduction

The properties of a material and its microstructure are closely related. In general, microstructural characterization involves the determination of crystalline structure, chemical composition, particle size and morphology, as well as phase distribution [1]. The traits present in the microstructure have different characteristics and require a large number of complementary techniques for their characterization [2].

Cement manufacturing and aggregates extraction for concrete production cause negative impacts on the environment. In light of this, investigations have analyzed alternative materials, which could partially replace the main constituents of concrete, since that they meet the requirements for acceptable structural, construction and safety performance. Also, they must present characteristics required by the current standards [3].

Self-Compacting Concrete (SCC) was developed in Japan by Okamura in 1986, with the intention to obtain a more durable concrete [4]. As a variation of high performance concrete (HPC), it presents exceptional deformability and has the ability to flow under its own weight, without the need of vibration. It also remains cohesive in the presence of dense reinforcement, and has high segregation resistance. A concrete mixture can only be considered as self-compacting if three following characteristics and fundamental properties are fulfilled to fresh concrete [5]:Filling Ability: Ability of SCC to flow into a mold and fill it completely only by its own weight;Passing Ability: Ability of SCC to flow through the steel reinforcing bars without segregation or blocking;Segregation Resistance: Ability of SCC to remain homogeneous during mixing, transport and placement.

SCC composition is the same of conventionally vibrated concrete (CVC), which consists of cement, fine and coarse aggregates, water, mineral and chemical admixtures. The most significant difference between the two concretes is that SCC requires less coarse aggregates and a higher quantity of fine materials, as well as a higher dosage of water reducing agents, superplasticizer admixtures (SP) and sometimes, viscosity modifying agents (VMA), to ensure adequate filling and passing abilities and segregation resistance [6]. Aggregate properties, particle size distribution, void ratio, slurry consumption and viscosity have a strong influence on SCC performance [7]. 

The use of cementitious materials such as blast furnace slag, microsilic, fly ash, metakaolin, limestone filler, rice husk ash, among others, has witnessed significant growth in recent years. This is due to the fact that these materials can provide cost reductions, decrease in environmental impact and pronounced improvement of the material final structural performance, such as in the case of compressive strength. In addition, some materials can help with issues of segregation and loss of workability, caused by overdose of superplasticizer [8].

Mineral additions are fine and inorganic materials incorporated into the concrete with intention to achieve or enhance specific properties. They can improve thermal cracking and reduce alkali-aggregate expansion, sulfate attacks and also, costs. The addition of these fine materials to the mixture, which are mostly industrial by-products, decreases their landfill disposal [9]. Multiple types of industrial waste have been used as alternative materials. In the scenario established in this paper, characterization techniques were necessary to verify if the studied material has desirable characteristics in the concrete production [3].

Chemical evaluation of the waste was performed by X-ray diffraction (XRD), X-ray fluorescence (XRF) and scanning electron microscopy (SEM) in energy dispersive spectroscopy (EDS) mode. Thermal analysis of thermogravimetry (TG) and differential scanning calorimetry (DSC) were used to determine if the material can assist in cement hydration processes and if pozzolanic properties are present [10]. These techniques also allow the evaluation of potential aggregates alkali-silica reactivity, which causes concrete expansion and cracking [11]. Opal, chalcedony and cristobalite, for example, can all be identified by XRD [12]. When the elements cannot be identifiable, other techniques such as Fourier transform infrared (FTIR) and differential scanning calorimetry (DSC) must be used [11]. According to brazilian standard NBR 12653 [13], pozzolanic material must comply with all physical and chemical requirements. These requirements include compositions greater than 70% in the sum of SiO_2_, Al_2_O_3_ and Fe_2_O_3_ percentages, a loss limit to flame of 6%, moisture content without exceeding 3%, and percentage of material retained in the 45 µm sieve less or equal to 34%.

Physical, chemical and mineralogical characteristics of the concrete constituent materials have a great influence on its mechanical properties [14], as well as its rheological and microstructural properties, both in fresh and hardened state. The more irregular is the particle, the less efficient is the grain packing, which has a significant influence in the aggregate void index, porosity and bulk density, being, in turn, altered by particle morphology, geometry and granulometric distribution [15]. 

Waste foundry sands (WFS) are a serious waste management problem for foundry industry, due to their high volume generated (from 75 to 80% of the total volume) [16]. Although sand recovery and regeneration processes are used for reuse in the foundry process [17], there is still a large volume of disposal related to a very fine-grained sand that cannot be reused [18,19] as, for example, the sand removed from exhaust systems. Almost all of this waste is landfilled and only 10% is used in other applications. The most likely reasons for this low utilization rate are that many engineers are unaware of the quality and properties of WFS, and some official associations are reluctant to use this by-product as a substitute material in various applications [17].

In industrial solid waste management, the measures are cleaner production programs, based on the principle of "anticipating and preventing" possible sources that generate environmental problems. These programs propose preventive, economic, environmental and technological strategies integrated with the processes and products, and aim to improve efficiency in the use of raw materials, water and energy, starting from non-generation, reduction and recycling of generated residues [20]. In this context, the reuse of discarded sand from silica-based foundry as aggregate for civil construction brings environmental benefits, such as reduction of carbon emission, CO_2_, by 20.000 tons. It generates savings of 20 million British Thermal Unit, BTU, of energy and 7.8 million liters of water [21]. Siddique et al. [22] stated that although the savings of CO_2_ emissions in atmosphere are not significant, the use of discarded foundry sand collaborates with the protection of environment. Turk et al. [23] studied the use of several materials derived from industrial processes (waste foundry sand, steel slag, fly ash and recycled aggregates) with respect to life cycle assessment (LCA). The authors, opus cit., confirmed 85% of reduction in environmental impacts with the use of WFS as a partial substitute for aggregate in conventional concrete. In addition, the use of concrete waste aggregate reduces the extraction of non-renewable mineral resources whose mining is responsible for approximately 1% of the total annual CO_2_ emissions (estimated as 4.1–10.8 million tons per year for portion of fine aggregate in concrete [21].

The consumption of natural resources in civil construction has been growing in recent decades. Given the current economic and environmental crisis, the search for a more sustainable environment has provided the development of the construction sector with new products as raw material for concrete [24]

The objective of this study is to characterize the material that cannot be regenerated, the waste foundry exhaust sand, using various techniques in order to analyze its constituents and verify if it can be used as a building material especially in SCC without impairing its strength and durability properties. In this way, the authors encourage the reuse of wastes materials in concrete increasing the possibilities for their application.

### Waste Foundry Exhaust Sand

Foundry waste industry generates numberless byproducts, ranging from used or waste foundry sand (UFS or WFS), slag, ashes, refractories, coagulant, exhaust system dust (filters), scrap, vapors and residual oven liquids [25,26]. Waste foundry exhaust sand (WFES) is a type of waste foundry sand. It is generated from the mixture of sand, bentonite and charcoal during the manufacturing of green sand molds for casting of metal parts. During demoulding of the parts on a vibrating conveyor, the waste is collected using a baghouse filter [27] in order to avoid particle suspension in the environment. Due to its reduced particle size, this waste is not regenerated, as it impairs the properties of molds, since it has limited permeability and makes it difficult for gases to escape. With this, bubbles are formed in the castings [28].

Physical and chemical characteristics of foundry sand depend on the type of process applied to the material, while classification depends on the type of binder used. The most commonly used are green and chemically bonded sands [29]. WFS, originated from green sand, is mainly made up of silica sand, carbonaceous additives to prevent sand fusion with metal, clay to increase cohesion of particles, and occasionally, metals, depending on the process. It is classified as non-hazardous, since it is not corrosive, ignitable, reactive or toxic and non-inert waste [30].

The characterization of foundry sands from molding, shot blasting and exhaust system sectors was carried out by Nyembwe et al. [31], to evaluate the viability of reusing these wastes in the construction industry. The authors concluded that the sand from molding sector can be used in concrete mixtures and for cement manufacturing. The finer sand, such as dust, can only be used for cement manufacturing, and shot blasting sand cannot be used due to the low silica and high metallic debris contents.

Exhaust powder is a residue taken from foundry exhaust systems, which has a smaller particle size than WFES. Ribeiro et al. [28] characterized this material for ceramic applications. X-ray fluorescence (XRF) technique was performed, and determined that the chemical composition was of 69.6% SiO_2_, 13.9% Al_2_O_3_ and 8.36% Fe_2_O_3_, among other components in quantities less than 1%. X-ray diffraction (XRD) also proved the presence of quartz in the crystalline phase (SiO_2_) and hematite (Fe_3_O_4_). Using scanning electron microscopy (SEM), grains of different sizes were observed with a predominance of 200 µm of particle size with rounded morphology. The results proved the formation of glassy structures, thus explaining the high resistance characteristics, low water absorption and no retraction. In his research, Ribeiro [32] concluded that steel residues such as WFS and exhaust powder could be used as ceramic materials for civil construction.

Partial cement replacement using foundry silica dust in self-compacting concrete was studied by Kraus et al. [33]. The average particle size (D50) for foundry silica powder was about 75 µm and approximately 74% thinner than 125 µm. This is a very important feature for SCC rheology, to prevent segregation. Scanning electron microscopy (SEM) showed that the waste presented an irregular morphology and a disordered surface. By X-ray diffraction (XRD), quartz (SiO_2_) was detected with traces of calcite (CaCO_3_), anortite (CaAl_2_Si_2_O_8_) and muscovite ((KAl_2_(Si_3_Al).O_10_(OH,F)_2_). Infrared spectroscopy (FTIR) showed a C-O band at 1453 cm^−1^, and an elevated presence of organic (22%), measured by loss on ignition. The authors concluded that foundry silica dust could be used to produce economically viable SCC.

Exhaust dust was characterized and its application in conventional concrete was investigated by Santos [34]. Particle size distribution analysis showed that 90% of the particles were smaller than 88.18 µm. Chemical composition by X-ray fluorescence showed 80.73% of SiO_2_ and in X-ray diffraction, a predominant presence of quartz was identified and it was confirmed by infrared spectroscopy. The authors concluded that it is possible to use the waste in conventional concrete without structural purposes.

Cúnico et al. [35], who found, by X-ray diffraction (XRD), the presence of SiO2 as silicate, analyzed exhaust powder as a ceramic coating. Also, montmorillonite from clay could be confirmed. The chemical composition found was of approximately 62% SiO_2_, 25% Al_2_O_3_ and 7% Fe_2_O_3_, in addition to potassium, calcium and titanium oxides in smaller proportions. For the material sintered at 1000°C, porosity was reduced due to better distribution of mixtures and decreased shrinkage.

Exhaust dust for application in construction industry was analyzed by Santos, Dalla Valentina, and Souza [36]. The authors (op. cit.) concluded that the residue has reduced particle size. It was observed that 90% of the particles were below 88.19 μm. By microstructural analysis performed using scanning electron microscopy (SEM), the authors observed that the residue presented an irregular lamellar morphology and in general, the grains were smaller than 180 µm. The mineralogical composition was analyzed using X-ray fluorescence (XRF). The images showed a predominant presence of silica oxide (SiO_2_), measured at 80.73%, in addition to 9.33% of aluminum oxide (Al_2_O_3_) and 4.01% of iron oxide (Fe_2_O_3_). Through X-ray diffraction (XRD) analysis, the presence of silica as quartz was confirmed. Using differential thermal analysis (DTA) and thermogravimetry (TG) analysis, the material was submitted to up to 1200 ºC. Total mass loss of 2.6% was observed, with the main decomposed components being aluminum oxide and iron oxide. Through Fourier infrared spectroscopy (FTIR) results, it was found that the peaks of 1090.59 cm^−1^, 798.34 cm^−1^ and 779.93 cm^−1^ were of silica oxide as quartz. The toxicological analysis showed that the waste does not present a risk to public health and to the environment. In view of the results, the authors concluded that the waste has characteristics similar to natural river sand.

Souza et al. [37] evaluated the possibility of using smelting exhaust powder to produce protective aluminum alloy coatings. X-ray fluorescence (XRF), scanning electron microscopy (SEM), energy dispersive spectroscopy (EDS) and X-ray diffraction (XRD) characterized the waste. The chemical composition identified by XRF was mainly of 83.13% of SiO_2_, 9.61% of Al_2_O_3_, and 4.73% of Fe_2_O_3_. The micrograph of exhaust dust seen using SEM showed that different morphologies and particle sizes compose it. Semi-quantitative analysis by EDS showed percentages by mass for each element, with 52.33% of oxygen, 15.53% of silicon, 12.76% of aluminum, and 3.21% of iron. XRD analysis identified quartz (SiO_2_), hematite (Fe_2_O_3_), potassium oxide (K_2_O), aluminium oxide (Al_2_O_3_), sodium oxide (Na_2_O_2_) and periclase (MgO). By FTIR spectrum, absorption bands were found in 1736 cm^−1^ and 1863 cm^−1^, and are related to the stretching of C=O groups. The bands corresponding to the wave numbers of 1528 cm^−1^ and 1863 cm^−1^ may be related to O-H bonds. Si-O bonds were observed in bands 980 cm^−1^ and 1182 cm^−1^ and may refer to quartz. The bands corresponding to Al-O were identified at 980 cm^−1^ and 798 cm^−1^, also for Al = O at 1080 cm^−1^ and for Al_2_O_3_ at 980 cm^−1^ and 684 cm^−1^.

Martins et al. [38] evaluated waste foundry exhaust sand as a fine aggregate for conventional concrete. Specific gravity was determined to be 2.61 g/cm^3^ with loose unit mass of 1418.17 kg/m^3^. Granulometric distribution was performed using a sequence of sieves. The particle size analysis indicated 50.59% of particles with size between 0.6 and 0.15 mm and 49.41% of particles smaller than 0.15 mm. By scanning electron microscopy (SEM), it was verified that the particles showed rounded morphology. The chemical elements present in the sample were identified by energy dispersive spectroscopy (EDS) as 55.77% oxygen, 22.45% silicon, 9.06% carbon, 5.99% aluminum, 2.52% sodium and 2.51% iron. Mineral composition was analyzed using X-ray diffraction (XRD), which identified the presence of silica quartz crystal (SiO_2_). As for toxicity, the waste was classified as non-hazardous and non-inert. The authors concluded that there was an increase in both compressive and splitting tensile strengths, and there was a decrease in absorption for mixtures with up to 50% of residue in relation to the control. Analysis of the leached and solubilized extract of the concrete with 50% of FES proved the waste inertization.

WFS was employed in the research of Bilal et al. [39] as a partial substitute (0%, 10%, 20%, 30% and 40%) for natural sand in the concrete. The authors evaluated workability, compressive, splitting tensile and flexural strengths, ultrasonic pulse velocity, Schmidt rebound hammer number and residual compressive strength. The tests were performed before and after exposure to high temperatures. The authors obtained maximum values for strength properties with a level of 30% of replacement.

In the research of Gurumoorthy and Arunachalam [18] fine aggregate was replaced by mass in percentages of 10%, 20%, 30% and 40% by treated used foundry sand (TUFS). The water absorption test was conducted on three specimens of each mixture. The samples were executed in the shape of cubes with 150 mm, following according to ASTM C 642-06 after 28 and 90 days of casting. Sulfate resistance test was conducted in accordance with ASTM C1012, being 10% by weight of Na_2_SO_4_ added to distillate water to produce a sodium sulfate solution. Test results indicated better performance for the concrete with TUFS, in comparison to control specimen. Also, the concrete with 30% of TUFS presented higher impermeability than control concrete, with better durability properties. 

Nithya et al [40] studied the properties of concrete containing waste foundry sand as partial replacement for fine aggregate in concrete, in different proportions (5%, 10%, 15%, 20%, 25%, 30%, 35 % and 40%). Compressive strength, split tensile strength, flexural strength and durability tests such as acid resistance and alkalinity measurement were conducted. There was a significant increase in mechanical strength of concrete with 35% of sand replacement. The authors affirmed that the waste can be successfully used as a building material, thereby minimizing or eliminating this environmental degradation. 

Self-compacting concrete with partial replacement of natural sand for foundry sand was analyzed by Gawande and Autade [41], in percentages of 0%, 10%, 20%, 30%, 40% by weight. The authors verified that workability and compressive strength at 7 days decreased as the percentage of WFS increases.

Siddique e Sandhu [42] evaluated the use of waste foundry sand as a partial substitute of natural sand in self-compacting concrete. Natural sand was replaced in four percentages (0%, 10%, 15%, 20%) of WFS by weight. Fresh properties of self-compacting concrete, compressive and splitting tensile strengths and durability tests were performed. Test results showed that there was an increase in compressive strength, with foundry sand (WFS) as partial replacement by sand up to 15%. Resistance of concrete against sulfates attack and rapid chloride permeability were also improved.

## 2. Materials and Methods

### 2.1. Waste Foundry Exhaust Sand Characterization

The sand in study comes from the exhaust system of the demoulding sector. This residue called waste foundry exhaust sand (WFES) is composed by quartz sand (99% SiO_2_), natural bentonite clay (hydrated aluminum and magnesium silicate), and light soda (sodium carbonate) to activate the bentonite and coal, Figure 1.

Physical and chemical of WFES properties were obtained in accordance with brazilian standards and by characterization techniques, as described in Table 1.

### 2.2. Self-Compacting Concrete Mixtures

In this experimental study, SCC mixtures were prepared using type V Portland Cement, silica fume as supplementary cementitious material (SCM), marble and granite waste as mineral addition, locally available natural sand and coarse aggregate with 9.5 mm of nominal size. Polycarboxylate ether based high-range water reducing admixture was used. Waste foundry exhaust sand (WFES) was obtained from a local foundry, and it was used as partial substitute for natural sand in percentages by weight of 0, 10, 20, 30 e 40 % (M0, M10, M20, M30 and M40, respectively). The mixtures designed for all specimens of SCC presented 455 kg/m^3^ of cement consumption and water to binder ratio of 0.35, as shown in Table 2.

### 2.3. Fresh and Hardened Concrete Testing Methods

Fresh properties of SCC were evaluated in accordance to NBR 15823-2 [53], NBR 15823-5 [54], NBR 15823-4 [55], brazilian standards that regulate the tests of filling ability, viscosity, passing ability and segregation resistance. The mixtures were subjected to slump flow and T500, V-funnel and L-box tests.

Compressive strength values were obtained, according to NBR 5739 [56]. Cylinders with Ø 100 × 200 mm were subjected to the test at 7 and 28 days. As durability indicators, water absorption and sulfate resistance parameters were analyzed. Water absorption test followed according to NBR 9778 [57], where cylinders of Ø 100 × 200 mm were evaluated at 30 days. Sulfate resistance test was based on ASTM C 452 [58], being used four cylinders with Ø 100 × 200 mm. The specimens were kept in magnesium sulfate, 5% MgSO_4,_ solution for 180 days, being after that submitted to compressive strength test.

## 3. Results

### 3.1. Waste Foundry Exhaust Sand Characterization

#### 3.1.1. Specific Gravity and Bulk Density

After analysis, the values of specific gravity and bulk density were determined as 2.62 g/cm^3^ and 1418 kg/m^3^, respectively. Authors such as Cúnico et al. [35] found in their study specific gravity of 2.2 g/cm^3^. Already Ashish et al. [59] obtained specific gravity of 2.64 g/cm^2^ and bulk density of 1820 kg/m^3^.

#### 3.1.2. Particle Size Distribution

The summary of particle size distribution of WFES by laser granulometry is indicated in Table 3. It can be seen that the residue showed reduced particle size, where 90% of the particles presented size below 211 µm. With this, WFES was classified as very fine sand.

From laser granulometry, the uniformity coefficient (UC) was obtained, which corresponds to the ratio between D60 (diameter corresponding to 60% in total weight of the particles smaller than it) and D10 (diameter corresponding to 10% in total weight of the particles smaller than it). According to NBR 7181 [60], the aggregate is considered uniform when UC is lower than 5. Particle size distribution was determined by laser granulometry, as shown in Figure 2.

Figure 2 shows the particle size distribution between 352.0 µm and 52.32 µm, with an average value close to 150 µm, which justifies the designation of the waste as exhaust sand and not exhaust dust since that exhaust dust has smaller granulometry then exhaust sand. This can be seen in the work of [36], where the residue presented low granulometry, with a high percentage of particles with size below 88 µm. Already in the study of [35], 90% of the particles showed size below 76.26 µm and mean diameter of 18.21 µm. The particles shape index and sphericity were evaluated using different particle sizes obtained from SEM images, with the aid of the Image -J 1.46r software [61] (Figure 3). 

Figure 3 are shown the tracings used to calculation of shape index, sphericity and of the f-circle of the grains. From the data obtained of specific gravity (SG), bulk density (BD) and particle size distribution, others physical characteristics as the maximum dimension characteristic (MDC), void index (e), porosity (η) and packing factor (E0) were calculated, as shown in Table 4.

According to NBR 7211 [62], sands are classified as coarse, medium, fine and very fine for the fineness modulus 3.20 < FM < 4.02, 2.40 < FM < 3.20, 2.0 < FM < 2.40 and FM < 2.0 respectively. Waste foundry exhaust sand is classified as very fine since FM < 2.0. As Angelin et al. [63] very fine sand has a porosity of 47%. The shape index ranged from 0.70 to 0.93 and the sphericity ranged from 0.88 to 0.97, with the averages of these values being adopted. According to Carasek et al. [64] 0.75 ≤ f-circle ≤ 1 indicates that the particles are circular or rounded. 

From laser particle size distribution, the uniformity coefficient (UC) was obtained, which corresponds to the relation between D60 (diameter corresponding to 60% in total weight of the particles smaller than it) and D10 (diameter corresponding to 10% in total weight of the particles smaller than it). According to NBR 7181 [60] the aggregate is considered uniform when UC is lower than 5.

About WFES physical properties and the SCC characteristics: Continuous particle size distribution, morphology (shape index), texture promote packing and rheological properties such as fluidity. In addition, the greater packing among particles requires lower paste volume and consequently, less cement. The internal porosity of particles also alters the density of hardened concrete, resulting in a more durable material, since that it is more difficult for aggressive agents to penetrate. The particles packing characteristics is of fundamental importance for the development of high-strength and high-performance concretes [65].

#### 3.1.3. Scanning Electron Microscopy (SEM)

Figure 4a,b show the micrographic images referring to the residue. The images were used to determine the grain sizes. The results of the SEM grain size distribution are consistent with the sieve analysis.

In Figure 4a the particles exhibited a certain homogeneity with rounded morphology. This classification was also determined by Martins et al. [38]. According to Santos, Dalla Valentina and Souza [36], exhaust powder has an irregular lamellar shape and in general, presents grains with sizes smaller than 180 µm. Already Parashar et al. [66] revealed that WFS particles have irregular shape. Figure 4b demonstrates the changes in the particles surface, that occur due to thermal shock that happens as a result of the powder contact with liquid metal during molding [28].

#### 3.1.4. Energy Dispersive Spectroscopy (EDS) 

Energy Dispersive Spectroscopy (EDS is the most used chemical analysis tool. is one of the widely used analytical methods for the analysis of qualitative information on elemental composition especially in conjunction with the use of a scanning electron microscope (SEM)

To quantify the chemical elements present in the sample, energy dispersive spectroscopy (EDS) was used. This analysis generated a spectrum with the elements identified in Figure 5 and weight percentages seen in Table 5.

The elements identified were silicon (Si) from quartz sand, in the form of aluminum silicate (Al), magnesium (Mg) and calcium (Ca) from bentonite clay and lastly, carbon and oxygen originated from coal and iron (Fe) during the smelting process. This result is close to that found in [37], who identified 52.33% of oxygen, 15.53% of silicon, 12.76% of aluminum, and 3.8% of iron, among other elements

#### 3.1.5. X-ray Diffraction (XRD)

X-ray diffraction (XRD) is a technique for characterizing crystalline materials. It provides information on structures, phases, preferred crystal orientations (texture), and other structural parameters. The XRD spectrum of WFES was obtained for diffraction angle 2θ ranged between 20° and 90° in steps 0.02. WFES mineralogical composition shown in Figure 6 indicates the presence of a single crystalline phase, identified as quartz (SiO_2_). Other elements present are contained in amorphous or crystalline phases in insufficient quantity to be detected by this technique. 

The predominance of silica (SiO_2_) was observed due to the composition of the raw material (green sand) used for molding castings. These results are compatible with those found in [36]. In Souza [67], quartz, hematite, potassium oxide, aluminum oxide, sodium oxide and periclase were identified. Parashar et al. [66] reported that WFS was mostly a crystalline material and presented quartz and albite as major phases.

#### 3.1.6. X-ray Fluorescence

X-ray fluorescence (XRF) has been used for the quali-quantitative evaluation of chemical elements. Table 6 shows the chemical composition analysis of the waste sand from the foundry exhaust. The main component seen is silicon oxide (81%) as quartz, confirmed using XRD, witch is a typical composition of natural sand.

This result corroborates with that presented in the XRD analysis. Similar results were found in [35] and [67]. For [35], it was observed a composition of 61.73% of SiO_2_, 25.37% of Al_2_O_3_, 37.38% of Fe_2_O_3_. Already for [67], a chemical composition of 83.12% of SiO_2_, 9.61% of Al_2_O_3_ and 4.73% of Fe_2_O_3_.

As for pozzolanicity, the sum of the compounds SiO_2_, Fe_2_O_3_ and Al_2_O_3_ was equal to 98.27% and therefore, higher than 70% as indicated in [13]. However, the fineness is not adequate for this material, which must have a retained percentage in the 45 µm mesh sieve of less than 34%. In this case, 100% was retained in the 45 µm mesh, thus concluding that this material cannot be considered as a supplementary cementitious material or a substitute for cement.

#### 3.1.7. TG/DTG Thermal Analysis

Thermal analysis is a grouping of techniques operates as a function of time and temperature to measure, mass variation (TG/DTG), temperature (DTA) and energy (DSC). It makes it possible to identify melting points, boiling points and phase diagrams, as well as patterns decomposition. Through Thermogravimetry (TG) technique, thermal decomposition, metal corrosion, calcination and other phenomena can be identified.

During thermogravimetry analysis, the residue was subjected to a temperature variation from 25° to 1000° C with a heating rate of 10°C/min. TG/DTG curves showed intervals associated with mass loss (Figure 7).

The waste showed a total mass loss around 6.1%, which was associated with decomposition of the constituents of free water residue and oxidation of organic matter, such as coal and disintegration of clay minerals. Santos, Dalla Valentina and Souza [36] state that the main decomposed constituents are aluminum oxide and iron oxide.

#### 3.1.8. Fourier Transform Infrared (FTIR)

Foundry exhaust sand was analyzed through infrared spectroscopy, in order to determine its chemical bonds. In the FTIR spectrum (Figure 8), peaks of 1080.33 cm^−1^, 796.67 cm^−1^ and 778.44 cm^−1^ could be seen, being correspondent to peaks of silica oxide (SiO_2_) in the form of quartz. The 693.61 cm^−1^ band was referent to Al_2_O_3_, and other peaks were not representative.

This result is very similar to that found by Santos, Dalla Valentina, and Souza [36]. In Souza [67], bands 980 and 1182 cm^−1^ were found referring to Si-O bonds and may refer to quartz. Santos [32] identifies bands corresponding to silicon in 1090, 798 and 779 cm^−1^. The bands corresponding to Al_2_O_3_ were identified at 980 and 684 cm^−1^ in [68].

#### 3.1.9. Waste Classification for Hazardousness 

Leaching and solubilization tests were performed and provided by the foundry industry. Table 7 and Table 8 show the main results.

Table 7 presents the concentration in the leachate extract obtained according to [50], of some chemical elements found in the waste which were compared to the maximum limits allowed in [52].

The parameters analyzed showed appropriated concentrations, according to those indicated in [52]. Thus, the residue was classified as non-hazardous.

Some elements solubilized in aqueous extract were obtained as described in [51] and the maximum limits established in [52] were compared as shown in Table 8.

With respect to the residue, the analyzed parameters of aluminum, barium, lead, total chrome, iron, manganese and nitrate exhibited higher concentrations than those presented in [52], which classifies the waste as non-inert. Based on the results obtained, the waste WFES was classified as Class II A—non-hazardous and non-inert. The sample showed a pH in water (1:1) of 9.51. Therefore, it was characterized as non-corrosive, according to the limit established in [52]. This result corroborates the result achieved in [36].

The applied techniques help to elucidate the chemical composition of materials. When applied together, these techniques provide a range of information that is extremely useful in identifying and characterizing materials, and can be complementary. About chemical properties and SCC characteristics: the composition of Waste Foundry Exhaust Sand is similar to natural sand and is classified as non-hazardous, non-inert and non-corrosive. The WFES mineralogical composition identified was quartz (SiO_2_). No chemical compounds have been identified to cause the alkali-silica reaction that causes expansion and cracking and consequently penetration of harmful agents.

### 3.2. Fresh SCC Properties

Results obtained for the properties in the fresh state can be seen in Table 9. All SCCs exhibited satisfactory slump flow in the range of 550–800 mm, which is an indication of a good deformability.

All mixtures presented the characteristics of SCC, such as fluidity, viscosity and segregation resistance. The mixtures showed increased spreading, T500 and V funnel time with an increase of up to 40% of WFES. These results are related to the flow tension and plastic viscosity of the mixtures, which are influenced by friction, shape and packaging between particles. As WFES has a more rounded morphology and a larger specific area than natural sand, it probably decreases flow stress and increase viscosity of the compositions. According to Castro et al. [65], the more spherical is the particle, the greater is the degree of packaging and the less is the friction, which facilitates the roll among grains.

### 3.3. Hardened SCC Properties

#### 3.3.1. Compressive Strength

The influence of WFES in the compressive strength of SCC mixtures (M0, M10, M20, M30, M40) at 7 and 28 days can be seen in Figure 9.

Continuous particle size distribution and spherical shape of the aggregates provided packing density between particles, which favored lower water demand and consequently, provided higher compressive strength with advancing ages for all mixtures. At 28 days, M30 mixture presented higher compressive values when compared to M0 mixture. This fact can be explained because the waste WFES is thinner than natural sand, which makes the composition denser. This result is similar to that obtained by Bilal et al. [39].

In Siddique et al [21] strength and durability properties of concrete improved with use of up to 20% of spent foundry sand (SFS) as a partial sand replacement. The thinner SFS material resulted in a denser concrete matrix and SFS concrete mixtures exhibited compressive strength up to 26% higher at 28 days. Siddique and Sandhu [42] concluded that the compressive strength of control concrete mixture was of 35.06 MPa at the age of 28 days. The percentage of increase in compressive strength was of 14.80, 22.73 and 14.26% for mixtures SCC-10, SCC-15 and SCC-20 in comparison to control mixture SCC-0. It was observed that compressive strength improved with the increase of WFS content up to 15% as partial replacement of sand

#### 3.3.2. Water Absorption and Voids

Figure 10 shows that there was a decrease in the void index and a consequent decrease in water absorption with 30% of residue in relation to the reference mixture.

It appears that the mixtures M0, M20, M30 could be classified in the category of low absorption and good quality, that is, absorption percentage lower than 3%, according to the comité européen du béton (CEB) [69]. The mixtures M10 and M40 fit as medium absorption and quality, since the percentage of absorption was between 3% and 5%, according to the classification proposed by CEB [69]. According to Helene [70], all concrete mixtures are classified as durable if the absorption rate is lower than 4.2%. These results can be attributed to the internal porosity, shape and packing of particles that improve the concrete density. Results obtained by Gurumoorthy, Arunachalam [18] showed that R30 mixture presented the lowest value at 28 and 90 days, indicating that 30% of replacement was considered very effective in reducing water absorption. 

#### 3.3.3. Sulfate Resistance

The resistance to magnesium sulfate attack after 180 days is shown in the Figure 11.

It was observed that there was an increase of 4.5% in compressive strength for M30 in comparison to control mixture after sulfate attack. These results can be attributed to internal porosity, shape and better packing of particles that improved the density of concrete, making it difficult for aggressive agents to penetrate. Gurumoorthy and Arunachalam [18] remarked that, in Na_2_SO_4_ solution, the minimum loss percentage in compressive strength was observed for R30, of 1.51% at 90 days. They affirmed that the use of treated used foundry sand (TUFS) as replacement material enhances the sulfate resistance of concrete. It was observed by Siddique and Sandhu [42] that for the mixture containing 10% of waste foundry sand, an increase in strength was verified at 56 days, after immersion in magnesium sulfate solution, when compared to control mixture. However, for both 15% and 20% of replacement levels, a decrease in strength was observed.

#### 3.3.4. Correlation between Voids Ratio and Sulfate Attack Resistance 

Sulfate attack resistance was related to water absorption of SCC mixtures. Correlation between these properties was verified using a linear equation, evaluating the coefficient R^2^ as shown in Figure 12.

The sulfate resistance curve showed R^2^ = 0.8403 for linear equations, indicating respectively, a high correlation between these properties.

The penetration of harmful agents in concrete does not only depend on the pore volume, but also on the connectivity between them. The results showed that, with 30% of sand replacement by WFES, SCC presented smaller void ratio and greater sulfate attack resistance. WFES fine particles acted as a packing material, and consequently, generated a concrete with denser matrix.

## 4. Conclusions

Self-compacting concrete is a relatively new material when compared to conventional vibrated concrete. The properties of fresh concrete such as fluidity, workability and viscosity depend very much on the characteristics and properties of its components.

Understanding the waste characteristics, it is possible to identify its influence on fresh and hardened concrete properties, such as compressive strength, water absorption, permeability and durability, shrinkage, and potential reactions when subjected to sudden changes in temperature.

Waste foundry exhaust sand is a low-particle waste with a chemical composition similar to natural sand and it is classified as non-hazardous, non-inert and non-corrosive. No chemical compounds that cause alkali-silica reaction were identified, which could lead to expansion and cracking, and consequently, penetration of harmful agents. WFES is considered an alternative material to replace natural sand in the construction industry.

The use of WFES as a partial replacement for fine aggregate in conventional concrete can increase water absorption and make the concrete less plastic. Therefore, its application is more suitable for SCC, since that it uses superplasticizer admixture, to increase fluidity, and fine materials, to improve viscosity and particles cohesion.

Due to its fineness, rounded shape and good packing between particles, WFES increases compressive strength, reduces water absorption and permeability of concrete mixtures and consequently, increases durability by making mixtures denser, thus reducing pore size and hindering penetration of harmful agents.

SCC mixtures using WFES as a partial substitute for natural sand presented appropriate required properties for fresh state such as fluidity, viscosity and segregation resistance. Good results were achieved for compressive strength and durability indicators. The best outcomes were obtained with 30% of residue.

This waste can be reused as mineral additive or fine aggregate in SCC. In this way, it can contribute to the development of sustainable concrete, since that it helps to reduce the use of cement and natural aggregates, and decreases the volume of waste in landfills.

## Figures and Tables

**Figure 1 materials-14-05629-f001:**
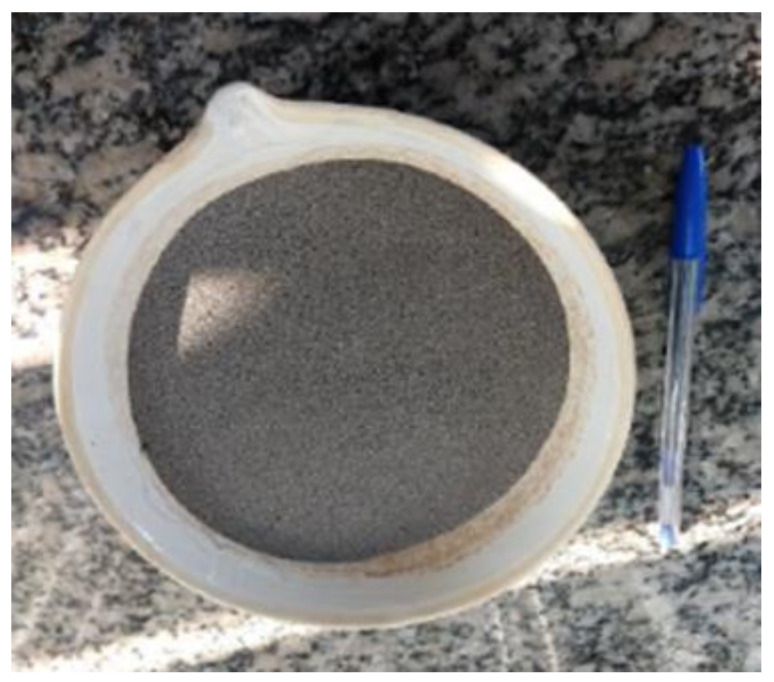
Waste foundry exhaust sand sample. Source: Authors.

**Figure 2 materials-14-05629-f002:**
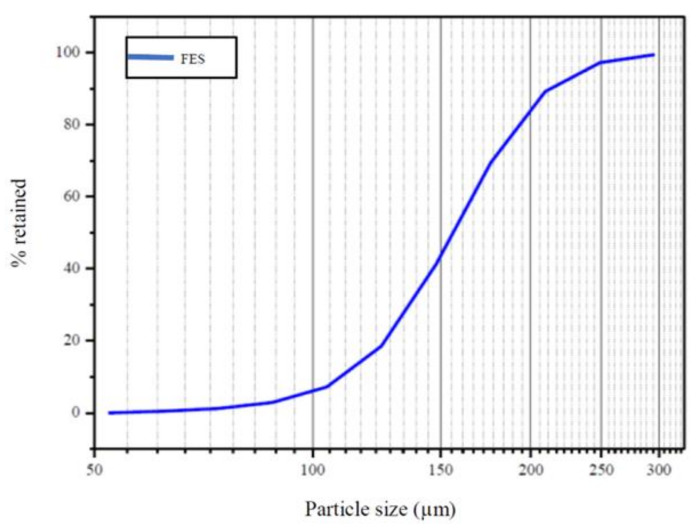
Waste foundry exhaust sand particle size distribution.

**Figure 3 materials-14-05629-f003:**
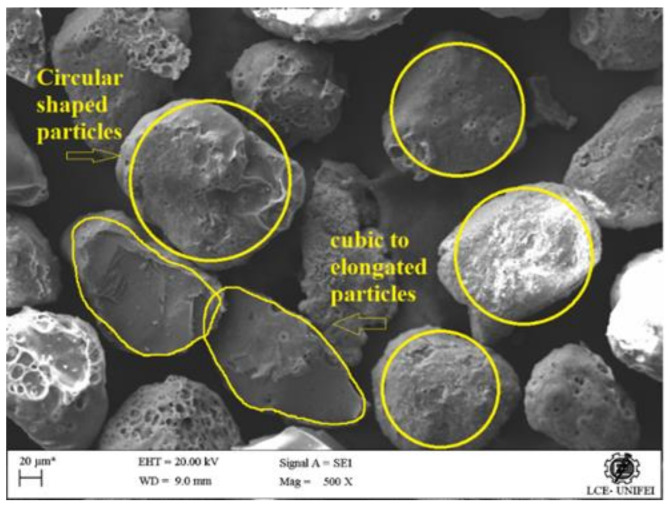
Waste foundry exhaust sand SEM micrograph, with details of shape and sphericity. Scale from 1 to 20 µm and magnification of 500×.

**Figure 4 materials-14-05629-f004:**
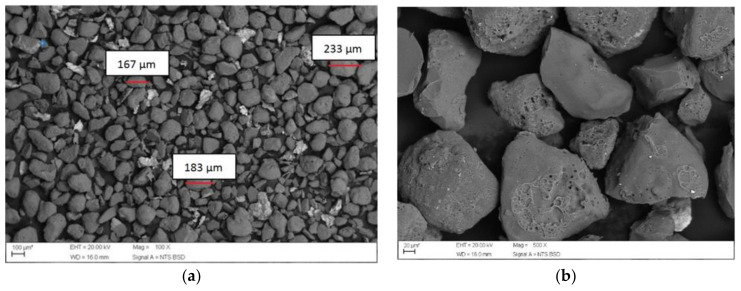
SEM by Back-Scattered Electron (BSE), (**a**) with 100× and (**b**) 500× magnification.

**Figure 5 materials-14-05629-f005:**
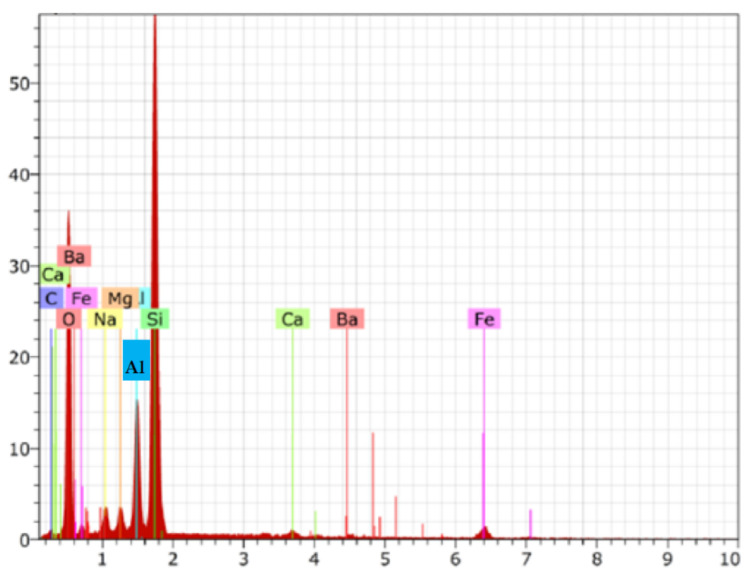
Spectrum obtained by microanalysis of X-rays characterized by EDS.

**Figure 6 materials-14-05629-f006:**
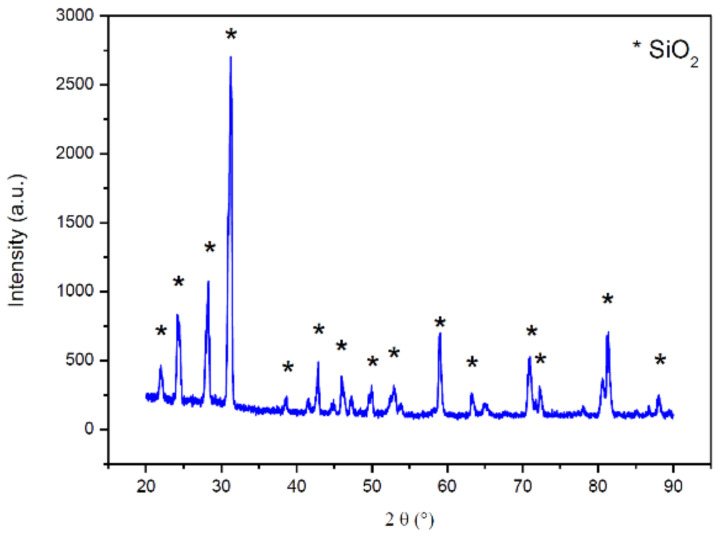
XRD diffraction of WFES mineralogical composition.

**Figure 7 materials-14-05629-f007:**
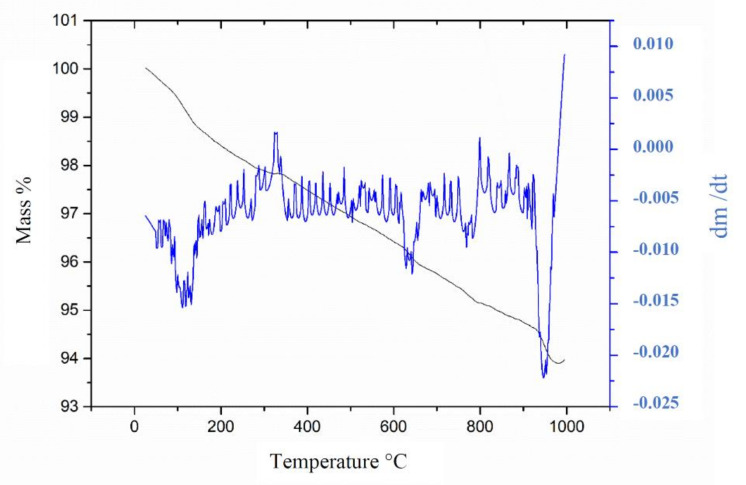
WFES TG and DTG curves.

**Figure 8 materials-14-05629-f008:**
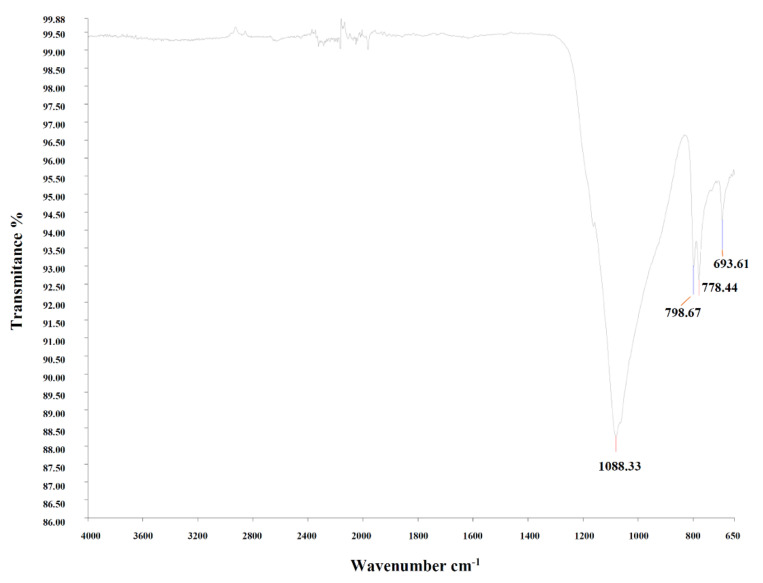
Fourier transform infrared spectra of WFES.

**Figure 9 materials-14-05629-f009:**
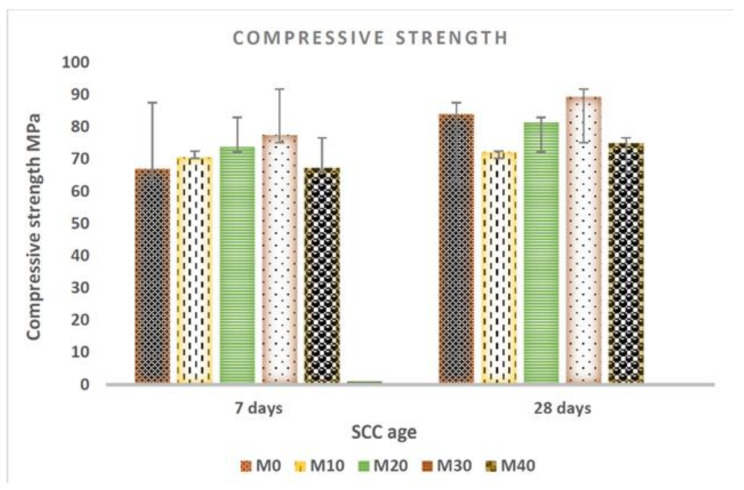
Compressive strength of SCC in different mixtures at 7 and 28 days.

**Figure 10 materials-14-05629-f010:**
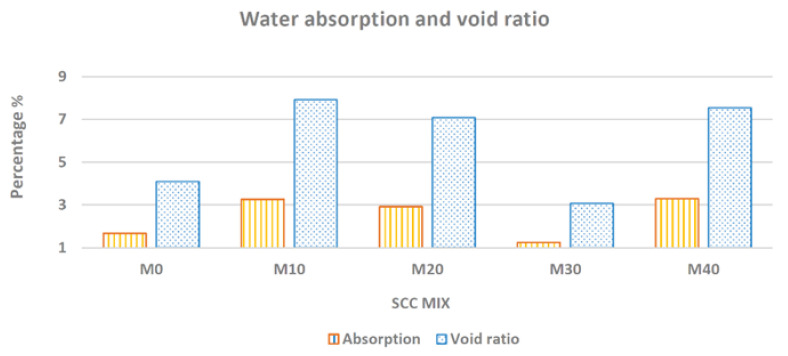
Water absorption and void ratio of different SCC mixtures.

**Figure 11 materials-14-05629-f011:**
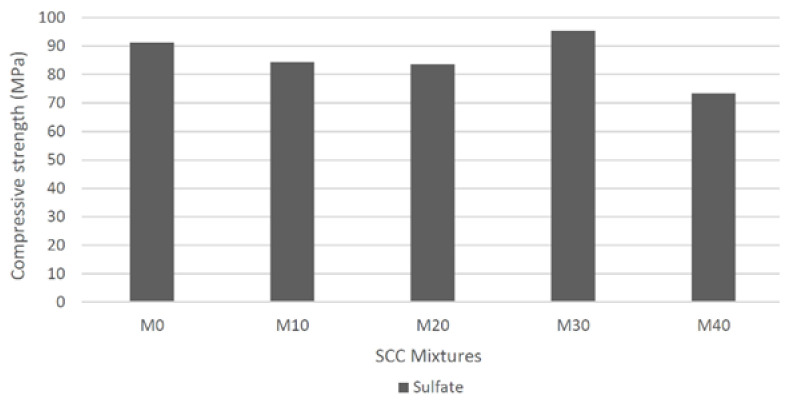
Compressive strength of SCC different mixtures after sulfate attack.

**Figure 12 materials-14-05629-f012:**
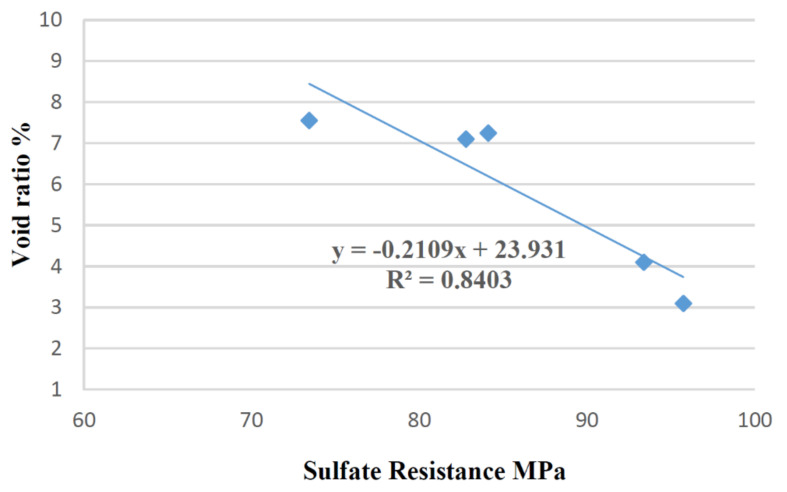
Correlation between voids ratio and sulfate attack resistance.

**Table 1 materials-14-05629-t001:** Characterization techniques applied to WFES waste.

Tests/Techniques	Analysis	ABNT Standard/Test Data	Equipment	Local
Specific gravity	-	NBR NM 52 [43]	Pycnometer/scale	Structures Laboratory—UNIFEI
Bulk density	Unit weight and air-void contents	NBR NM 45 [44]		
Particle size analyser Laser diffraction	Granulometry	-	Microtrac, S 3500 [45].	Structural characterization laboratory—UNIFEI
Scanning electron microscopy (SEM)	Morphology/chemical elements	Backscattered Electrons (BSE)/ (energy dispersive spectroscopy (EDS)	Zeiss^®^ Model EVO MA-15. [46]	
X-ray diffraction (XRD)	Mineralogical phases	Scan between 20° a 90° 2 Theta.	PANalytical^®^, model Xpert-Pró [47]	
Thermogravimetry (TG) and derivative thermogravimetry (DTG)	Physical and chemical reactions due to temperature variation	Temperature from 25 ° C to 1000 ° C and rate of 10 ° C/min	Metler MT 15 [48]	Biomaterials Laboratory—UNIFEI
Fourier transform infrared spectroscopy (FTIR)	Atomic structure	Sweep from 450 cm^−1^ to 40,000 cm^−1^	Perkin Elmer, Model Spectrum 100 [49]	Chemistry Laboratory—UNIFEI
X-ray fluorescence (FRX)	Chemical composition	Fusion after lost on ignition	Axios MAX, PANalytical [47]	USP Lorena
Leached extract	Toxicity and waste classification	NBR 10005: 2004 [50]		TASQA^®^ Analytical Services Ltd., Paulínia, SP
Solubilized extract		NBR 10006: 2004 [51]		
Waste classification		NBR 10004: 2004 [52]		

**Table 2 materials-14-05629-t002:** SCC mixtures.

Mix	Cement	SF	MGPW	WFES	Sand	Coarse	SP	Water	W/Binder
M0	1	0.06	0.300	0.000	1.896	1.626	0.008	0.37	0.35
M10	1	0.06	0.300	0.189	1.707	1.626	0.008	0.37	0.35
M20	1	0.06	0.300	0.378	1.517	1.626	0.008	0.37	0.35
M30	1	0.06	0.300	0.569	1.328	1.626	0.008	0.37	0.35
M40	1	0.06	0.300	0.759	1.138	1.626	0.008	0.37	0.35

**Table 3 materials-14-05629-t003:** WFES particle size distribution.

	Particle Size Distribution	Size (µm)
D_10_	10% of the particles below	110.9
D_30_	30% of the particles below	137.1
D_50_	50% of the particles below	156.1
D_60_	60% of the particles below	165.8
D_90_	90% of the particles below	211.7
D_95_	95% of the particles below	232.1
D_100_	100% of the particles below	352.0

**Table 4 materials-14-05629-t004:** Characteristics determined for WFES.

Physical properties	SG g/cm^3^	Bulk Density g/cm^3^	MDC mm	FM	e	η %	E_0_	SI	Φ mm/mm	f-Circle	UC
WFES	2.62	1.418	0.3	0.51	0.85	0.46	0.54	0.79	0.91	0.94	1.49

MDC (maximum dimension characteristic), e (voids ratio) = SG/(BD-1); η (Porosity) = [e/(1+e).100]; E_0_ (Packaging factor) = 100-η; SI (shape index) = D2/D1; Φ (Sphericity) = (4Ap/ℼ)^1/2^.(1/D1); f-circle = 4ℼAp/P^2^; UC (Uniformity Coefficient) = D60/D10. Where: D1 = Diameter of smallest circumference that circumscribes the grain (mm); D2 = Diameter of the largest circumference inscribed in the grain (mm); Ap = Projected grain area (mm^2^/mm^2^); P = Grain perimeter (mm).

**Table 5 materials-14-05629-t005:** Elements analyzed by EDS.

Element	Atomic Number	Standard Weight %
O	8	51.59
Si	14	26.16
Al	13	7.72
C	6	5.32
Fe	26	3.20
Na	11	2.60
Mg	12	1.82
Ca	20	0.58
Ba	56	0.41
Total		100.00

**Table 6 materials-14-05629-t006:** WFES chemical composition.

Chemical Compost	Concentration %
SiO_2_	81.08
Fe_2_O_3_	15.97
Al_2_O_3_	1.22
Na_2_O	0.40
MgO	0.38
CaO	0.20
MnO	0.12
TiO_2_	0.11
P_2_O_5_	0.09
K_2_O	0.09
CuO	0.09
SO_3_	0.07
NiO	0.06
Cr_2_O_3_	0.05
ZrO_2_	0.04
Nb_2_O_5_	0.02
MoO_3_	0.01

**Table 7 materials-14-05629-t007:** Analysis of leachate extract from WFES.

Parameter	Analytical Results mg/L	Maximum Limit mg/L
Arsenic	<0.05	1
Barium	0.75	70
Cadmium	<0.005	0.5
Lead	<0.028	5
Total chrome	<0.005	1
Fluorides	0.13	150
Mercury	<0.00017	0.1
Selenium	<0.005	1
Organic	<LQ	F Attachment

**Table 8 materials-14-05629-t008:** Analysis of WFES solubilized extract.

Parameter	Analytical Results mg/L	Maximum Limit mg/L
Aluminum	16.4	0.2
Arsenic	0.006	0.01
Barium	3.94	0.7
Cadmium	<0.005	0.005
Lead	0.013	0.01
Cianeto	0.0064	0.07
Chloride	40.3	250
Copper	0.019	2
Total chrome	0.13	0.05
Iron	10.9	0.32
Fluorides	0.79	1.5
Manganese	0.47	0.1
Mercury	0.0002	0.001
Nitrate (as N)	19.0	10
Silver	<0.005	0.05
Selenium	<0.002	0.01
Sodium	79.5	200
Sulfate (SO_4_)	101	250
Surfactants	<0.03	0.5
Zinc	0.021	5
Phenols	<0.0059	0.01

**Table 9 materials-14-05629-t009:** Rheological properties of SCC mixtures.

Mixtures	Slump Flow	T_500_	V Funnel	L Box
	mm	sec	sec	H1/H2
M0	780	3	5	0.96
M10	780	2	4	0.92
M20	785	3	7	0.96
M30	800	3	7	0.97
M40	800	6	12	0.97

## Data Availability

The data presented in this study are available in the article.

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
