# Peer review of "Physical and Chemical Properties of Waste Foundry Exhaust Sand for Use in Self-Compacting Concrete"

_materials, 2021, doi:10.3390/ma14195629_

Round 1
Reviewer 1 Report
This paper could be made effective with minor revision.
Please gives more details about sand, What about fineness modulus of sand?
Figure 10. Compressive strength in SCC mixtures...Please correct the title of Fig. 10. Compressive strength of SCC with different mixtures.
You have to choose one equation only in Figure 13.
The authors of this paper missed a lot of articles on this topic. A suitable references to that article should be included in the paper.
- Self compacting concrete with partial replacement of sand by waste foundry sand..International Journal of Advance Engineering and Research
Development Volume 4, Issue 9, September -2017 - Properties of Self-Compacting Concrete Incorporating Waste Foundry
Sand...Leonardo Journal of Sciences, Issue 23, July-December 2013
p. 105-124 - Properties of concrete containing waste foundry sand for partial replacement of fine aggregate in concrete... Article in Indian Journal of Engineering and Materials Sciences · April 2017
Author Response
Reviewer #1:
- Please gives more details about sand, What about fineness modulus of sand?
The by-products generated by the foundry industry are foundry slag, sand, iron scrap, particulate matter, exhaust dust and furnace fumes. Foundry exhaust sand originates from the manufacturing process of sand molds and during demolding of metal parts. It is a fine sand rich in silica in the form of quartz, collected by filters (baghouse). This sand is discarded because it makes it difficult for gases to escape during molding due to its high content of fines, harming the foundries.
The fineness modulus of the WFES is 0.51 as shown in Table 3.
- Figure 10. Compressive strength in SCC mixtures...Please correct the title of Fig. 10. Compressive strength of SCC with different mixtures.
In the new version of the article, the authors eliminated de Figure 5 so Figure 10 changed to Figure 9. The title of Figure 9 (Figure10) was corrected.
- You have to choose one equation only in Figure
The authors eliminated de Figure 5 so Figure 13 changed to Figure 12. The authors chose de linear equation.
- The authors of this paper missed a lot of articles on this topic. A suitable reference to that article should be included in the paper.
- Self compacting concrete with partial replacement of sand by waste foundry sand. International Journal of Advance Engineering and Research Development Volume 4, Issue 9, September -2017
- Properties of Self-Compacting Concrete Incorporating Waste Foundry Sand...Leonardo Journal of Sciences, Issue 23, July-December 2013 p. 105-124
- Properties of concrete containing waste foundry sand for partial replacement of fine aggregate in concrete... Article in Indian Journal of Engineering and Materials Sciences • April 2017
In agreement with the suggestion of the reviewer, the authors included the indicated references.

Reviewer 2 Report
In this paper, the purpose of this study is to characterize and analyze Waste Foundry Exhaust Sand (WFES) for use in Self-Compacting Concrete (SCC).
On the whole, this study provides a good reference for the WFES application. A series of experiments have been done. Authors carried out many microstructure tests, however, the intrinsic connection or relationship between these results haven’t been well analyzed. The results just describe the experimental results but lack of deep analysis. And almost all the figures’ name are too short to show the meaning clearly. So, before the article is published, these problems should be solved. And the other problems are as follows.
Comment 1: Page 1 Abstract: The abstract part has less description of the conclusion, and it is suggested to supplement about the waste.
Comment 2: page 2 Line 55: Delete “and”
Comment 3: page 3 Line 116: How much discarded sand can reduce the corresponding carbon dioxide, and how much discarded sand can save the corresponding energy? Here, the corresponding relationship of data is fuzzy.
Comment 4: Figure 4 and Figure 5 have the same name. Please figure out the details and differences between them in the figure’s name.
Comment 5: page 12 Figure 6: The chemical elements represented by the blue line are blocked, please adjust it.
Comment 6: Page 12 Line 370: Figure 7 is not properly described in this section. For example, its peak corresponds to the abscissa.
Comment 7: Page 15 Figure 9: The data in the figure is too small to be clear.
Comment 8: All table styles in the article are not unified.
Comment 9: page 19 Figure 13: There are few data in the figure, so it's better to supplement it.
Author Response
Reviewer #2:
In this paper, the purpose of this study is to characterize and analyze Waste Foundry Exhaust Sand (WFES) for use in Self-Compacting Concrete (SCC).
On the whole, this study provides a good reference for the WFES application. A series of experiments have been done. Authors carried out many microstructure tests, however, the intrinsic connection or relationship between these results haven’t been well analyzed. The results just describe the experimental results but lack of deep analysis. And almost all the figures’ name are too short to show the meaning clearly. So, before the article is published, these problems should be solved. And the other problems are as follows.
The authors worked in the organization of the article. The discussion of results was restructured.
… the intrinsic connection or relationship between these results haven’t been well analyzed
The applied techniques help to elucidate the chemical composition of materials. When applied together, these techniques provide a range of information that is extremely useful in identifying and characterizing materials, and can be complementary. About chemical properties and SCC characteristics: the composition of Waste Foundry Exhaust Sand is similar to natural sand and is classified as non-hazardous, non-inert and non-corrosive. The WFES mineralogical composition identified was quartz (SiO2).No chemical compounds have been identified to cause the alkali-silica reaction that causes expansion and cracking and consequently penetration of harmful agents
Comment 1: Page 1 Abstract: The abstract part has less description of the conclusion, and it is suggested to supplement about the waste.
The authors completed the abstract as the reviewer suggested.
Comment 2: page 2 Line 55: Delete “and”
It was done.
Comment 3: page 3 Line 116: How much discarded sand can reduce the corresponding carbon dioxide, and how much discarded sand can save the corresponding energy? Here, the corresponding relationship of data is fuzzy.
… In this context, the reuse of discarded sand from silica-based foundry as aggregate for civil construction brings environmental benefits such as the reduction of carbon emission, CO2, by 20.000 tons. It generates savings of 20 million BTU of energy and 7.8 million liters of water [21]. Siddique et al. [22] stated that although the savings in CO2 calls in the atmosphere is not significant, the use of discarded foundry sand collaborates with the protection of the environment. Turk et al. [23] studied the use of several materials derived from industrial processes (waste foundry sand, steel slag, fly ash and recycled aggregates) with respect to Life Cycle Assessment (LCA). The authors, opus cit., confirm the 85% reduction in environmental impacts with the use of WFS as a partial substitute for aggregates in conventional concrete. In addition, the use of concrete waste aggregates reduces the extraction of non-renewable mineral resources whose mining is responsible for approximately 1% of the total annual CO2 emissions (estimated as 4.1–10.8 million tons per year for the fine aggregate portion in concrete [24].
Comment 4: Figure 4 and Figure 5 have the same name. Please figure out the details and differences between them in the figure’s name.
Figures 4 and 5 represented the same properties so it were grouped and its name was changed to Figures 4a and 4b.
Comment 5: page 12 Figure 6: The chemical elements represented by the blue line are blocked, please adjust it.
In the new version of the article, the authors eliminated de “Figure 5”. Therefore, Figure 6 changed to Figure 5.The chemical elements represented by the blue line is Aluminium in Figure 5. It was adjusted.
Comment 6: Page 12 Line 370: Figure 7 is not properly described in this section. For example, its peak corresponds to the abscissa.
As already explained, the authors eliminated de “Figure 5”. Therefore, Figure 7 was changed to Figure 6. About the peaks in figure 6: All identified peaks correspond to SiO2.
Comment 7: Page 15 Figure 9: The data in the figure is too small to be clear.
As already explained the authors eliminated de “Figure 5”. Therefore, Figure 9 was changed to Figure 8. The figure was reformulated.
Comment 8: All table styles in the article are not unified.
Actually several mistakes were identified on the tables. The authors reviewed all of them.
Comment 9: page 19 Figure 13: There are few data in the figure, so it's better to supplement it.
As already explained the authors eliminated de “Figure 5”. Therefore, Figure 13 was changed to Figure 12. Unfortunately, there is no more data to add to this analysis but the authors sought to deepen the discussion of the result.
.
The manuscript has now been resubmitted to your journal. We look forward to hearing from you and would like to thank you for all your input so that we may successfully publish our manuscript. We remain at your disposal for any further corrections.
Yours sincerely,
The authors.
